

# Anthropogenic climate change has increased severity of mid-latitude storms and impacted airport operations

Lia Rapella[1,2], Tommaso Alberti[3], Davide Faranda[2,4], and Philippe Drobinski[1]

[1]LMD-IPSL, Ecole Polytechnique, Institut Polytechnique de Paris, ENS, Université PSL, Sorbonne Université, CNRS, Palaiseau, France
[2]Laboratoire des Sciences du Climat et de l'Environnement, UMR 8212 CEA-CNRS-UVSQ, Université Paris-Saclay & IPSL, CE Saclay l'Orme des Merisiers, 91191, Gif-sur-Yvette, France
[3]Istituto Nazionale di Geofisica e Vulcanologia, via di Vigna Murata 605, 00143 Rome, Italy
[4]London Mathematical Laboratory, 8 Margravine Gardens, London, W6 8RH, British Islands

**Correspondence:** Lia Rapella (lia.rapella@lmd.ipsl.fr)

**Abstract.** The impact of extreme weather events, particularly those associated with tropical and extra-tropical cyclones (TC and ETC), on aviation can rise serious concerns in the context of the ongoing climate change. These events often lead to significant disruptions, including flight cancellations, delays, re-routing, and impacts on airport infrastructure resilience to adverse weather conditions. This study conducts an analysis of the influence of anthropogenic climate change on four recent

major storm events that occurred over Europe, the USA, and East Asia, with an in-depth analysis on the Storm Eunice, a powerful ETC that affected the UK and Ireland. Using climate reanalysis data we assess the dynamics of these extreme storms and their implications for aviation operations, particularly during critical phases such as take-off and landing. Our research underscores the growing intensity of extreme storms, particularly stronger winds, driven by human-induced climate change, and stresses the need for taking into account growing climate hazards to optimize planes and airport operations.

## 1 Introduction

The Sixth Assessment Report (AR6) of the Intergovernmental Panel on Climate Change (IPCC) states that anthropogenic climate change is intensifying extreme weather events (Masson-Delmotte et al., 2021). Global warming is altering atmospheric patterns, leading to an increase in the frequency and severity of extreme events (Archer and Caldeira, 2008; Marvel and Bonfils, 2013). Among these events, thunderstorms and strong winds associated with tropical and extra-tropical cyclones (TC and

ETC) are particularly impactful (Cheung and Chu, 2023). For the aviation sector, they pose significant risks both considering on-ground and in-flight effects, enhancing episodes of both Clear Air Turbulence (CAT) and Convective-Induced Turbulence (CIT). These events affect fuel efficiency, passenger comfort, and operational safety, particularly in the lower atmosphere where most commercial aircraft operate (Sharman and Lane, 2016). Specifically, CIT has been estimated to account for 60% of turbulence-related aircraft accidents (Cornman and Carmichael, 1993). Storms can also create unfavorable weather conditions

at airports, leading to flight cancellations, delays and, consequently, economic losses (International Civil Aviation Organization, 2018). For instance, visibility issues due to rain or fog can hinder take-off, while strong wind shear can alter the lift an airplane



experiences during take-off. According to the Federal Aviation Administration guidelines (Federal Aviation Administration, 2022), landing or taking off should be avoided during an approaching storm, as sudden gust fronts and low level turbulence can lead to a loss of control (Williams, 2014).

Several studies have highlighted the effects of adverse weather on worldwide airport and airline operations. For example, Robinson (1989) found that over 165000 minutes of delays per year at Atlanta Hartsfield International Airport were attributable to adverse weather. Changnon (1996) noted that increased rainfall in the late 1970s led to more departures with delays over 30 minutes at Chicago O'Hare Airport. Sasse and Hauf (2003) reported that thunderstorms significantly raised delays at Frankfurt Airport by a factor of 6.3 in 1997 and 1.1 in 1998. Hsiao and Hansen (2006) observed that average delays on days with adverse

weather conditions were 14 minutes longer than on clear days. Borsky and Unterberger (2019), covering 10 large US airports, concluded that flights which face a weather shock are delayed by up to 23 min. In the Republic of Korea, Kim et al. (2023) showed that rainfall had a greater impact on aircraft cancellations compared with wind speed. Oo and Oo (2022) found that thunderstorm rain poses the largest risk for aviation operation at the Yangon International Airport in Myanmar.

    However, under the ongoing global warming these impacts can be exacerbated by changes in the characteristics of TC and

ETC, potentially increasing the risks for aviation, altering take-off and landing distances, as well as climb angles (Gratton et al., 2022). The influence of climate change on extratropical cyclones have been addressed by numerous studies (e.g., Zappa et al., 2013; Ulbrich et al., 2009; Priestley and Catto, 2022). The IPCC report (Lee et al., 2021) suggests that, while the overall intensity and number of ETC may not change, the associated precipitation and impacts will likely increase, particularly in the North Atlantic during winter, although with regional variations. Similarly, tornadoes, particularly those associated with

land-falling tropical cyclones, may become more common in the USA (Wu et al., 2022). These tornado outbreaks could become more dangerous due to increased frequency, duration, and a higher likelihood of occurring at night (Forbis et al., 2024). Understanding these changes is crucial for the aviation industry, as it navigates the evolving risks posed by a changing climate. Several recent studies have provided evidence of increasing turbulence levels and episodes, especially at mid-latitudes (Williams and Joshi, 2013; Williams, 2017; Storer et al., 2017, 2019; Alberti et al., 2024). Nevertheless, attribution-oriented

studies for examining the link between climate change and storms and how this link can affect aviation operations are still missing. These studies can be helpful for stakeholders to be better prepared for the challenges ahead and implement strategies to mitigate the associated risks. These risks are particularly relevant in regions such as Europe, the USA, and Asia which have dense air traffic networks and significant economic activities (Burbidge, 2023). In this context, the framework of attribution science (Otto, 2016, 2019; Yiou et al., 2017) represents the most powerful tool for assessing changes in the frequency and

spatial patterns of specific extreme events, as well as the probability that climate change influences these events (e.g., Faranda et al., 2023; Faranda et al., 2024).

    The present study aims to propose a way to detect the influence of anthropogenic climate change on four notable storms events associated with ETC/TC that occurred over Europe, the USA, and East Asia between 2022 and 2023. Our aim is to offer a first observation-based framework to attribute impacts of extreme events in aviation to greenhouse gases emissions.

In particular, we focus on storm Eunice, a powerful ETC that mainly affected UK and Ireland in February 2022, providing a detailed analysis to illustrate the methodology. Additionally, we examine storm Poly, a severe windstorm impacting northwest





Europe in early July 2023; a strong North American windstorm, responsible for widespread impacts across much of the USA in late February 2023; and Typhoon Hinnamnor, which affected the western Pacific in late August 2022. We decided to focus on these events since they had significant impacts on aviation, resulting in numerous flight cancellations and delays. In our

analysis, we assess atmospheric variables at ground level since our focus is on issues that could prevent aircraft from taking off and/or disrupt landings, as well as, forcing flight cancellations and airports' closures, rather than on challenges that may arise during flight as in previous studies (e.g., Alberti et al., 2024). The paper is organized as follows. In section 2, we present the data and the methods which we will use in our forthcoming analysis. In section 3 we first present detailed results for storm Eunice, followed by main results for the other events. We draw our conclusion in section 4.

## 2 Data & Methods

### 2.1 ERA5 reanalysis data

For our analysis, we use the ERA5 reanalysis dataset (Hersbach et al., 2023), which has a spatial resolution of $0.25°\times0.25°$, covering the period from 1950 to 2023, at 6-hourly temporal resolution. Specifically, we use different variables in our analysis: the mean sea level pressure, SLP [hPa], the geopotential height at 500 hPa, Z500 [m], the 2-m air temperature, T [°C], the total

precipitation, TP [mm day$^{-1}$], and the 10-m wind speed, $V$ [m s$^{-1}$] to characterize the event from a meteorological point of view. Additionally, the velocity field at the two pressure levels closest to the ground, namely 1000 hPa and 975 hPa, is used to quantify its impacts on airport operations, not only in terms of adverse weather conditions featured by airports' infrastructures but also during take-off and landing operations. To the latter purpose, we evaluate the velocity field gradient as it plays a crucial role in generating the lift required for an aircraft to take off, together with turbulence-related metrics, i.e., the Ellrod's

indices TI1 and TI2, operationally developed to assist in forecasting CAT and potentially CIT in the atmosphere (Ellrod and Knapp, 1992). TI1 [s$^{-1}$] is the product of vertical wind shear and horizontal deformation, providing a measure of turbulence generation due to wind shear-induced strain. TI2 [s$^{-1}$] incorporates the development of frontal zones, accounting for both dynamic and thermodynamic properties of the atmosphere, and is particularly useful for identifying regions prone to aviation turbulence (Ellrod and Knox, 2010). In both cases, larger values of TI1 or TI2 are usually associated with areas of increased

potential for CAT or CIT development. As an additional metric, we also evaluated a proxy for the eddy dissipation rate (EDR) based on ERA5 data (Alberti et al., 2024). The EDR quantifies the rate at which turbulent kinetic energy is transferred down to smaller scales and ultimately dissipated as heat due to viscous effects (International Civil Aviation Organization, 2018). This metric provides insight into the energy cascade process in turbulence, reflecting the intensity of mixing and the efficiency of kinetic energy conversion into thermal energy within the atmosphere. EDR is widely used to classify turbulence into different

classes/levels (International Civil Aviation Organization, 2018): when EDR≤ 0.2, turbulence is light, moderate turbulence occurs when 0.2 <EDR≤ 0.45, while severe turbulence is experienced when EDR> 0.45.




## 2.2 Method

In this section we present the procedure we use to detect the influence of climate change on ETCs. To detect recurring patterns in large-scale weather events we follow the methodology presented by Faranda et al. (2024), based on the analogues methodology outlined by Yiou et al. (2017); Ginesta et al. (2024). Each event is represented through its synoptic conditions via the SLP (or the Z500) field over a specific spatial domain that contains the low-pressure area associated with the storm/cyclone[1]. We designate as the *cyclone time* the time-step corresponding to the SLP minimum over this area. Then, the SLP field of the event is compared with all others in the database to find its weather analogues, i.e., a sample of events minimizing their Euclidean distances with the event itself (Faranda et al., 2024).

The full database is then divided into two non-overlapping periods of equal duration and long enough for statistical characterization of extreme events, natural variability but short enough to assume that extreme events have similar characteristics in each period. We choose 35-years periods – as customary in attribution studies (Luu et al., 2018; Vautard et al., 2019) – to strike a balance between ensuring the climate state remains relatively stable and being long enough to smooth out short-term fluctuations in atmospheric dynamics. The less recent period [1950–1984] is representative of climate conditions less impacted by global warming, usually defined as *factual period*; the most recent period [1989–2023] reflects a stronger anthropogenic influence (Gulev et al., 2021), usually defined as *counterfactual period*. For both factual and counterfactual periods, we select the thirty best analogues. As is typical for attribution studies, the cyclone itself is not included in the list, and analogues overlapping over a 7-day consecutive period are also excluded. Furthermore, we search analogues only over the months of the extended season during which the cyclone time occurred, i.e., December-January-February-March (DJFM) as the winter season and June-July-August-September (JJAS) as the summer season. This accounts for potential shifts in atmospheric circulation due to climate change and to restrict the analysis to events typical of a specific season, which are more likely to present similar patterns and characteristics (Faranda et al., 2024). This also prevents mixing the distinct physical processes that contribute to storms in warm versus cold seasons.

For each period, we compute SLP, Z500 and T anomalies over the time-steps corresponding to the identified analogues. Specifically, we removed at each grid-point and for each time-step the average of SLP, Z500 and T values for all the corresponding calendar values in the period 1950-2023. We then calculate a weighted average of the anomalies over the two time windows. The weights are calculated as the reciprocal of the logarithm of the Euclidean distance of each specific analogue, which gives more importance to the analogues closer to the event. Finally, we take the difference between the two averages, to assess whether changes occurred between a climate with and without climate change. The analysis extends to impact-oriented variables, i.e., TP, V, TI1, TI2, and EDR. For these variables, composite maps are generated directly through weighted logarithmic averaging of the data, along with the difference maps between factual and counterfactual scenarios. To test the significance of the differences, we apply the bootstrap method at the $95^{th}$ level of confidence (Faranda et al., 2024). Finally, to account for the potential influence of low-frequency modes of natural variability, we use the El Niño-Southern Oscillation (ENSO), the Atlantic Multi-decadal Oscillation (AMO) and the Pacific Decadal Oscillation (PDO) monthly indices values. We applied

---

[1]The method is robust with respect to changes in the domain size, provided that the storm/cyclone is fully contained in the domain (Faranda et al., 2024).





a two-sided Kolmogorov-Smirnov test to assess the significance of changes in the distributions of analogues between the two periods. If the p-value is below 0.05, we reject the null hypothesis, indicating distinct distributions. If low-frequency variability is excluded, changes in analogues between present and past period are attributed to climate change. We are aware that we are not testing all possible mode of variability but as discussed in Faranda et al. (2024) and Fery and Faranda (2024) the main influences on the meteorological time scales come from these three modes. Moreover, we also examine changes in the monthly

occurrence of analogs to determine whether there is a shift in atmospheric circulation within the season.

## 3   Results

We present a detailed case study for the Storm Eunice to provide a comprehensive overview of the methodology. For the other events, we will directly focus on the impacts at airports. Additional figures can be found in the Appendix A.

### 3.1   Storm Eunice

Storm Eunice, the second of three storms that affected the UK in February 2022, brought severe weather across the UK and Ireland, causing fatalities and significant transport disruptions (Kendon, 2022). A significant disruption in air traffic has led to the cancellation of more than 400 flights across the UK, with the majority involving Heathrow Airport (Timmins, 2022). In Ireland, Dublin Airport was also heavily impacted, with a large number of both incoming and outgoing flights being can-

celed (The Sun, 2022). European insurance costs were estimated to be around 2.5 billions euros (Rosanes, 2022). Initially a weak system around 40°N, 50°W at 06:00 UTC on February 16, Eunice rapidly intensified over the next 24 hours, deepening explosively to a central pressure of 975 hPa, positioned southwest of Ireland, by 00:00 UTC on February 18. At its peak, 11:00 UTC on February 18, the storm produced a record-breaking wind gust of 55 m/s at the Needles Old Battery on the Isle of Wight. Surface wind speeds reached 28 m/s in the English Channel and remained strong across land, with most stations record-

ing speeds over 15 m/s. This intensity was partially attributed to a *sting jet*, a narrow air stream within the storm capable of producing exceptionally strong winds over localized areas (Volonté et al., 2024a). This phenomenon occurs in some cyclones classified as Shapiro-Keyser. For a detailed explanation of this phenomenon during storm Eunice, we refer to Volonté et al. (2024b).

As our focus is on impacts on airports disruptions, we have selected the British Islands as the spatial domain for the attribution

analysis, given the significant disruptions experienced in this region (Kendon, 2022). This area fully contains the low pressure area associated with the cyclone, with the cyclone time identified at 06:00 UTC on 18 February 2022, when the storm's center was located just south of Ireland. This can be easily located by the minimum in the SLP anomaly (Figure 1**a**), showing a wide depression up to −40 hPa. At that time, Eunice was still in its early stages, reaching its mature phase only around 11:00 on the same day, while crossing southern UK before continuing its track eastward. During these early stages, the sting jet remained

strong over the area, clearly visible in satellite images (see Figure 3 in Volonté et al., 2024a). This is also consistent with the Z500 and T anomaly patterns displaying a strong positive gradients of approximately 500 m and 12 °C, respectively, between





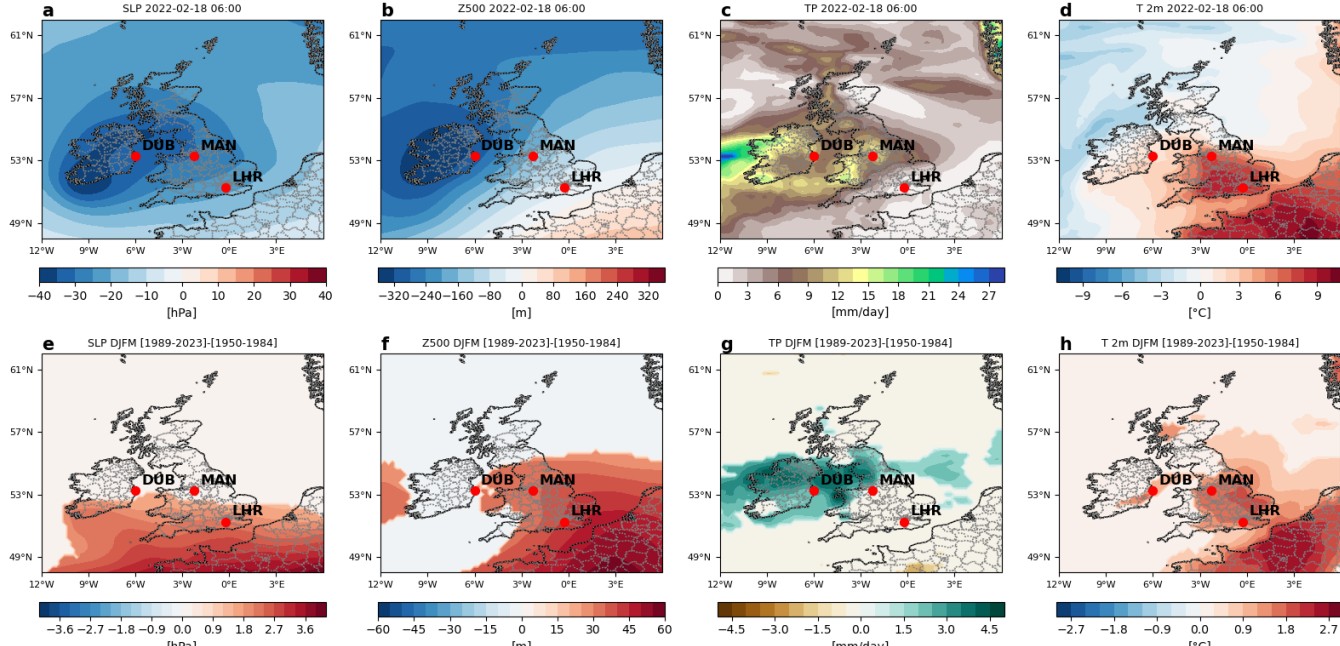

**Figure 1.** Analogue-based results for the Storm Eunice. SLP **a** anomaly, Z500 **b** anomaly, TP **c** and 2-m T **d** anomaly at the cyclone time. Difference between factual [1989-2023] and counterfactual [1950-1984] period of the average anomalies at the time-steps corresponding to the analogues for SLP **e**, Z500 **f**, TP **g** and 2-m temperature **h**. In the second row, shadings indicate significant changes. Red dots indicate major airports in the region: Dublin Airport (DUB), Manchester Airport (MAN) and Heathrow Airport (LHR).

the area around the cyclone center and northern France (Figure 1**b,d**). At the cyclone time, the heaviest rainfall was observed over western Ireland and Wales (Figure 1**c**), with peak values reaching up to 27 mm/day.

Results by comparing past and present period suggest that this type of cyclones are now shallower than in the past (Figure
1**e**), showing increased anomalies up to ∼4 hPa over continental Europe and up to ∼1 hPa close to the center of the cyclone. Similarly, increased geopotential height anomalies (up to 50-60 m) and air temperatures anomalies (up to 2.7 °C) are also observed in the present compared to the past (Figure 1**f,h**), although no changes are observed close to the cyclone eye. This could indicate a north-east shift of the cyclone core in the present climate, suggesting changes in the storm trajectory. Nevertheless, present-day cyclones bring more precipitation (up to 4.5 mm/day) with respect to the past (Figure 1**g**), significantly higher
upstream of the center, over Ireland and few areas in central UK.

Looking at impacts on wind speed variations, we can observe that during the event the strongest winds were concentrated below the cyclone center, reaching up to 30 m/s (Figure 2**a**) on ground and up to 40 m/s (Figure 2**b**) at 975 hPa (around 400 m). Furthermore, present-day cyclones are characterized by higher-than-normal wind speeds, with anomalies of up to 2 m/s on ground (Figure 2**c**) and 4 m/s at 975 hPa (Figure 2**d**), particularly over regions such as the southern UK and the Dublin area.
These wind speed increases align with the Z500 anomaly pattern observed during the event and in the analogues scenarios.



Specifically, a gradient in Z500 anomalies at the cyclone time was detected between the area around the cyclone center and northern France, with pressure lines becoming narrower toward the southwest part of the domain (Figure 1**b**). In the present climate, the same areas are characterized by a steeper gradient in geopotential height, as suggested by the larger Z500 values over the high-pressure region, corresponding to increased wind speed intensity. At the cyclone time these areas experienced

wind speeds exceeding 7 m/s, a critical threshold for aviation disturbances (International Civil Aviation Organization, 2008). This suggests that more extensive regions during Eunice-like storms in the present climate will be exposed to winds surpassing this threshold, amplifying challenges for aviation operations. In fact, these conditions favor stronger vertical wind shear that could potentially affect aviation operations, especially during take-off or landing phases, creating crosswinds or tailwinds. The

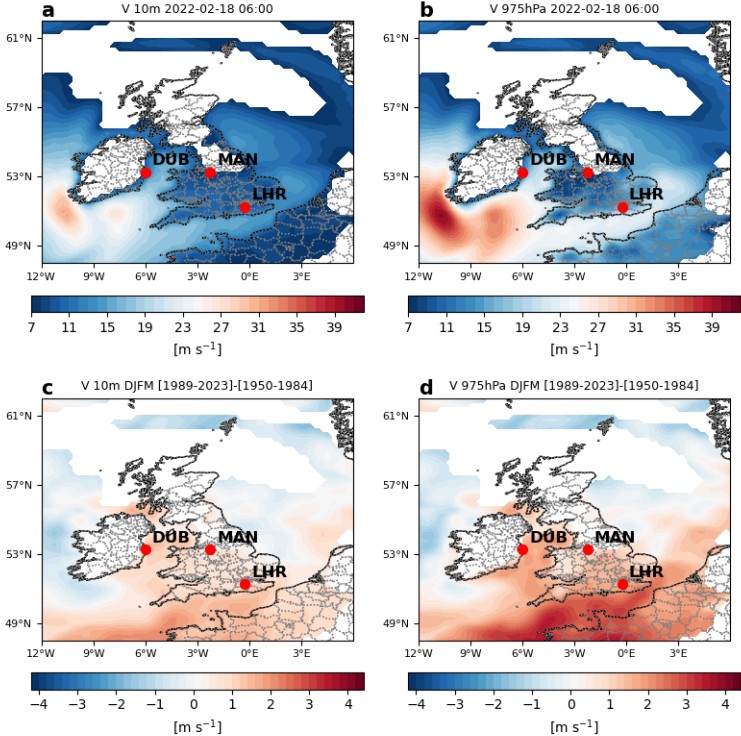

**Figure 2.** Analogue-based results on the wind speed for the Storm Eunice. 10-m **a** and 975 hPa **b** wind speed at the cyclone time. Difference between factual [1989-2023] and counterfactual [1950-1984] period of the average field at the time-steps corresponding to the analogues for 10-m **c** and 975 hPa **d** wind speed, respectively. Red dots indicate major airports in the region: Dublin Airport (DUB), Manchester Airport (MAN) and Heathrow Airport (LHR).

storm Eunice generated stronger (up to 11 m/s) vertical wind speed gradients (Figure 3**a**) in a region further south than the

center of the cyclone, as well as over the English Channel, the southern coasts of the UK and the northern French coasts. Over the same regions larger values of all turbulence-related metrics, i.e., TI1, TI2, and EDR, are observed (Figure 3**b-d**), thus suggesting enhanced risks for coastal airports during take-off or landing. By comparing present episodes with past events we



observe increased velocity shears up to 2 m/s over the sea and Northern France (Figure 3**e**), with localised increases in TI1 over

Southern coasts of Ireland, Eastern and Southern coasts of UK, and Northern coasts of France (Figure 3**f**). Larger increases

in TI2 (Figure 3**g**) are observed over Northern French coasts and Eastern coasts of UK, while an overall increase in the EDR

(Figure 3**h**) is observed over almost the Southern part of the analyzed domain. None of the modes considered to analyze the

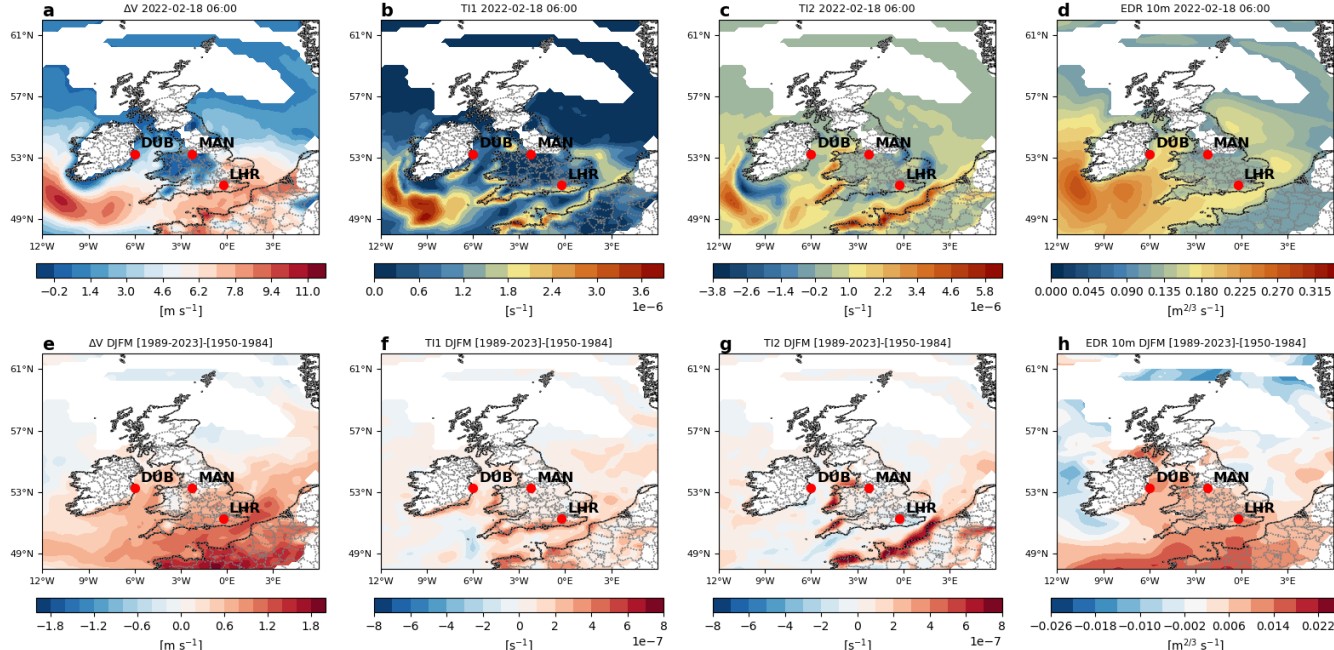

**Figure 3.** Analogue-based results on aviation-related metrics for the Storm Eunice. Vertical wind speed difference $\Delta V = V1000hPa - V975hPa$

**a**, Ellrod index TI1 **b**, Ellrod index TI2 **c**, and EDR **d** at the cyclone time. Difference between factual [1989-2023] and counterfactual [1950-

1984] period of the average composites at the time-steps corresponding to the analogues for vertical wind speed difference **e**, Ellrod index

TI1 **f**, Ellrod index TI2 **g**, and EDR **h**. White areas indicate regions with V at the cyclone time lower than 7 m/s. Red dots indicate major

airports in the region: Dublin Airport (DUB), Manchester Airport (MAN) and Heathrow Airport (LHR).

influence of natural climate variability (ENSO, AMO, PDO) shows significant changes in their distribution for the analogues

between the counterfactual and the factual periods over the considered extended season (Figure 4**a-c**). This suggests that the

observed changes are likely due to the effects of climate change. Within the DJFM season, the analogues frequency is higher

in December and March, while decreasing during the colder months of January and February (Figure 4**d**).

The analysis of Storm Eunice indicates a noticeable shift towards December for Eunice-like storms in the current climate.

Eunice-like storms are not only becoming more frequent at the beginning of the winter season, but also more severe in a warm-

ing climate, likely due to the impacts of climate change. Indeed, the potential influences of ENSO, AMO and PDO have been

ruled out as significant factors in these changes. These findings align with the results of Ginesta et al. (2024), who also ob-

served an increase in precipitation and wind severity associated with Eunice-like storm analogues in future climate scenarios.




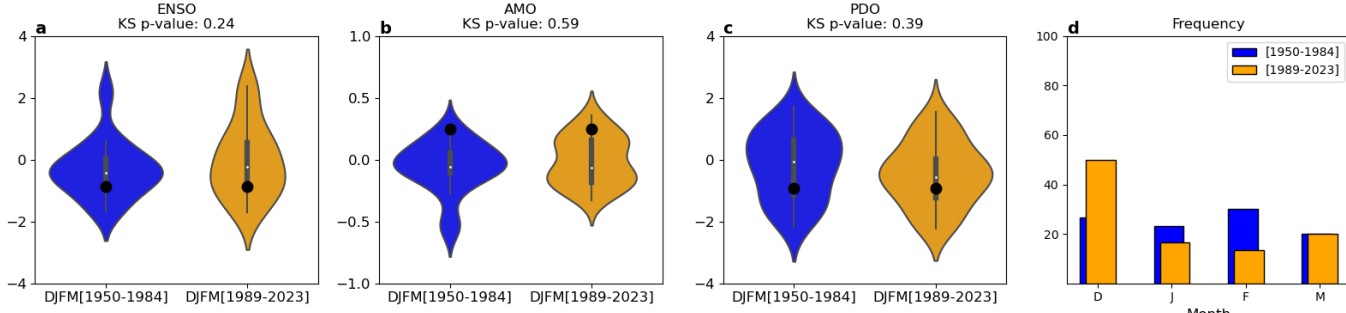

**Figure 4.** Violin plots showing the distribution of monthly index values for ENSO **a**, AMO **b**, and PDO **c** during the past (blue) and present (yellow) periods. Values for the peak day of the extreme event are marked by a black dot. Panel **d**: distribution of the frequency (%) of analogues occurring in each sub-period.

However, their study focused on mature stages of the cyclone, thus analyzing analogues with respect to a different cyclone time and over a different area. When combined with our findings, this supports the conclusion that climate change is playing a key role in the intensification of these storms throughout their stages.

## 3.2 Storm Poly, USA windstorm on February 2023, and Typhoon Hinnamnor

As mentioned in section 1, besides storm Eunice, we applied the analogue-based methodology to three other events, associated with ETC/TC, responsible for huge impacts on aviation, whose results are reported below.

Storm Poly, an exceptionally intense summer ETC, that struck Germany, the Netherlands, and Denmark on July 5, 2023, caused extensive damage, and severe air travel disruptions. With hurricane-force gusts reaching 146 km/h, it became the strongest summer storm ever recorded in the Netherlands. The storm developed rapidly over the North Atlantic on July 4 and intensified over the North Sea, bringing heavy rain and strong winds, particularly to the Netherlands, where over 400 flights were canceled at Amsterdam Airport due to extreme conditions. The SLP anomalies (Figure A1**a**) show a large negative anomaly, up to $-20$ hPa over the coast nearby Amsterdam, with Z500 anomalies (Figure A1**b**) displaying up to $-250$ m over the same area. Precipitation data (Figure A1**c**) show high daily amounts of precipitation over the Amsterdam region and along the Norwegian coast, reaching up to 20 mm/day. Temperature data (Figure A1**d**) show negative anomalies over the coast, from Amsterdam to the Copenhagen, while showing positive anomalies in the mainland of Germany (up to 4 °C). Wind speed data (Figure A1**e**) indicates strong winds over the coast nearby Amsterdam. These winds enhanced precipitation over the same areas contributing to Amsterdam Airport disruptions. When analyzing how meteorological conditions similar to those that led to Storm Poly have changed in the present period compared to how they occurred in the past, we observe a slightly deeper pressure ($-1$ hPa, Figure A1**f**), with shallower geopotential heights up to 50 m (Figure A1**g**), leading to increased precipitations over the sea up to 6 mm/day (Figure A1**h**), with higher temperature gradients (Figure A1**i**) over the coast (+ 2°C) and the sea



(+ 1°C), and stronger winds over the coast up to 2 m/s (Figure A1**j**). Beyond these changes, we detect the influence of natural climate variability on the event, specifically both the ENSO and AMO tele-connection patterns (Figure A2**a-b**), while no influence is detected from PDO (Figure A2**c**). This means that the changes we see in the events compared to the past are also

215 partially driven by natural variability. Furthermore, we observe that similar events are now more probable to occur in January and February, while previously mostly occurring in February and March (Figure A2**d**).

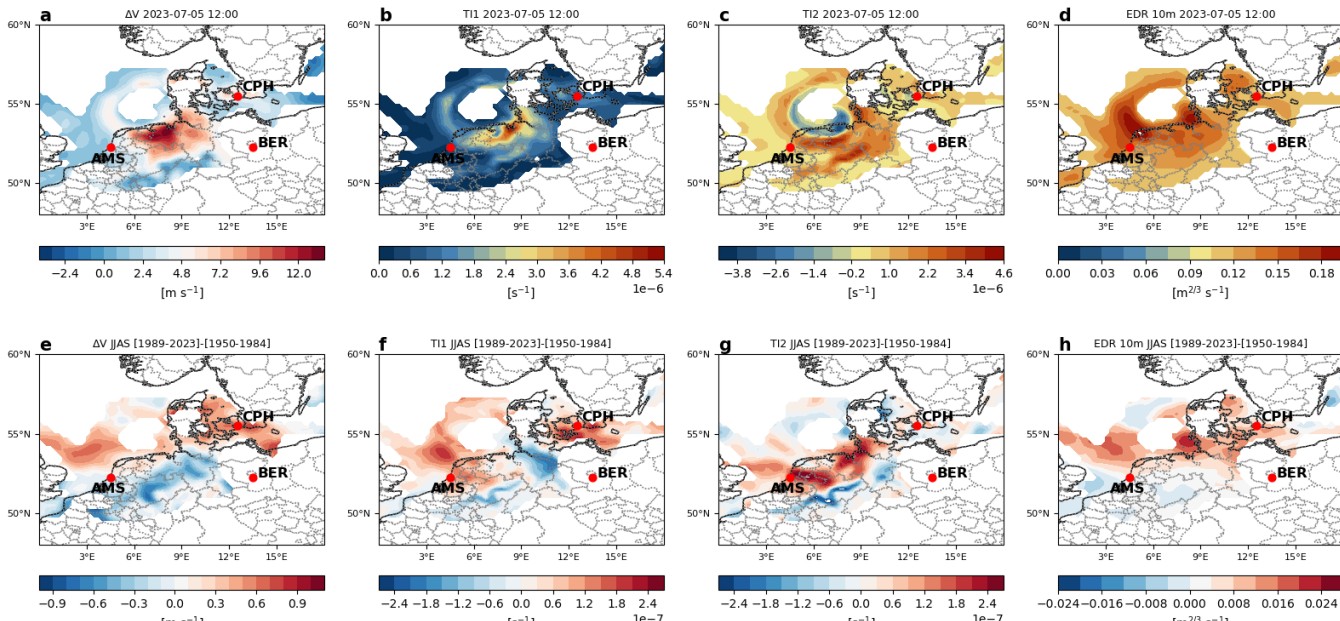

**Figure 5.** Analogue-based results on aviation-related metrics for the Storm Poly. Vertical wind speed difference **a**, Ellrod index TI1 **b**, Ellrod index TI2 **c**, and EDR **d** at the cyclone time. Difference between factual [1989-2023] and counterfactual [1950-1984] period of the average composites at the time-steps corresponding to the analogues for vertical wind speed difference **e**, Ellrod index TI1 **f**, Ellrod index TI2 **g**, and EDR **h**. White areas indicate regions with V at the cyclone time lower than 7 m/s. Red dots indicate major airports in the region: Amsterdam Airport Shipol (AMS), Copenhagen Airport (CPH) and Berlin Brandenburg Airport (BER).

A strong North American windstorm brought heavy snow, freezing rain and strong winds in February 2023, causing flight disruptions and leading to numerous cancellations and delays at major airports, especially in the Midwest and Northeast. The SLP anomalies (Figure A3**a**) show a large negative anomaly, up to $-40$ hPa, in a region between Las Vegas and Denver,

220 with Z500 anomalies (Figure A3**b**) displaying up to $-320$ m slightly westerly with respect to the same area. Precipitation data (Figure A3**c**) show extremely high daily amounts of precipitation over the same region, reaching up to 30 mm/day. Temperature data (Figure A3**d**) show negative temperature anomalies over the inner regions. Wind speed data (Figure A3**e**) indicates strong winds easterly to the cyclone center up to 18 m/s. A comparison of meteorological conditions similar to those that led to the North American windstorm between the present and the past reveals that no significant changes in pressure (Figure A3**f**) and




geopotential heights (Figure A3**g**) occurred. We do not observe increased precipitations, apart some northward regions where up to 4 mm/day increases are detected (Figure A3**h**), with no changes in temperature (Figure A3**i**), while stronger winds, close to the cyclone center, are observed up to 2 m/s (Figure A3**j**). Beyond these changes, we do not detect any influence of natural tele-connection patterns (Figure A4**a-c**), suggesting that, as with Storm Eunice, the detected changes are likely due to anthropogenic climate change. Additionally, we do not detect any significant changes in their seasonal occurrence, apart a slight increase in the occurrence of similar events in February while previously occurring mostly in January (Figure A4**d**).

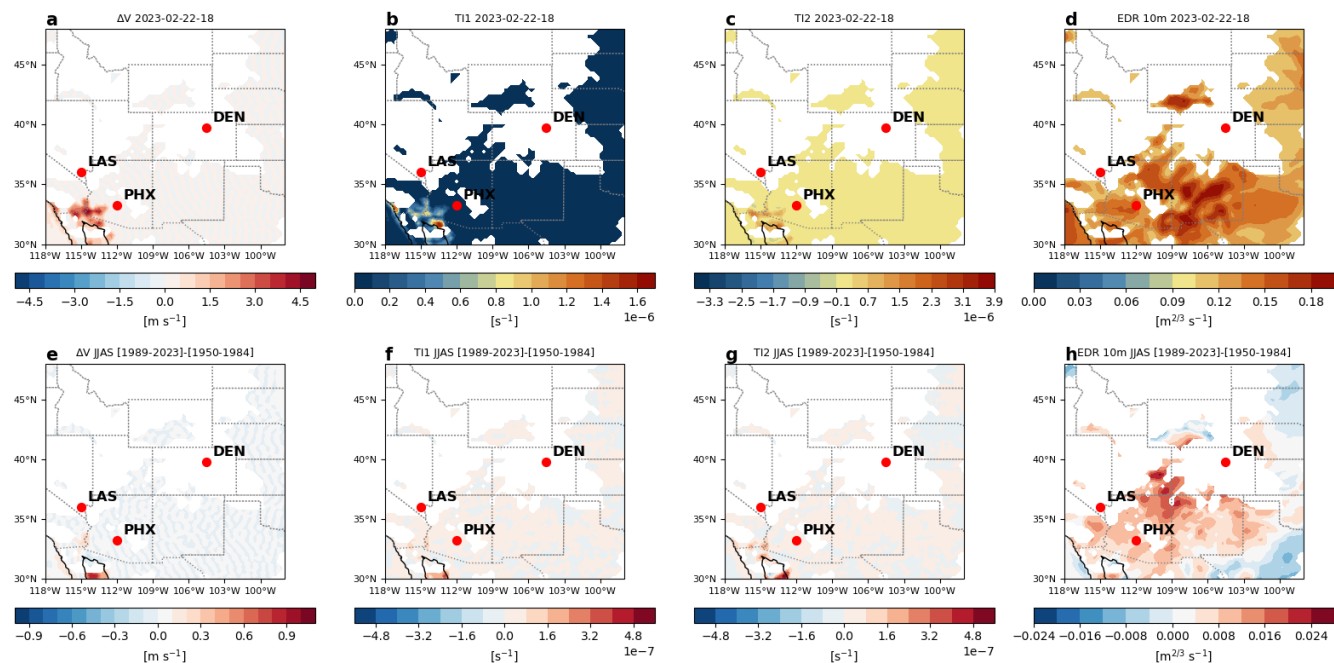

**Figure 6.** Analogue-based results on aviation-related metrics for the 2023 February North American windstorm. Vertical wind speed difference **a**, Ellrod index TI1 **b**, Ellrod index TI2 **c**, and EDR **d** at the cyclone time. Difference between factual [1989-2023] and counterfactual [1950-1984] period of the average composites at the time-steps corresponding to the analogues for vertical wind speed difference **e**, Ellrod index TI1 **f**, Ellrod index TI2 **g**, and EDR **h**. White areas indicate regions with V at the cyclone time lower than 7 m/s. Red dots indicate major airports in the region: McCarran International Airport (LAS), Phoenix Sky Harbor International Airport (PHX) and Denver International Airport (DEN).

Typhoon Hinnamnor, a powerful and destructive TC, impacted East Asia in late August/early September 2022 (Kim et al., 2024). Hinnamnor caused significant disruption, including power outages, and damage to infrastructure, due to high winds and heavy rain. Airports such as Gimhae International in South Korea and Kansai International in Japan saw widespread delays and closures. The SLP anomalies (Figure A5**a**) show a large negative anomaly, up to $-20$ hPa southerly to Japan, where the cyclone eye is located, with associated Z500 anomalies (Figure A5**b**) up to $-200$ m over the same area. Precipitation data (Figure A5**c**) show very extremely high daily amounts of precipitation in the southern-easterly region, reaching up to 180 mm/day.



Temperature data (Figure A5**d**) show very extremely high temperature anomalies (up to 7 °C) in the southern-easterly region, while no significant changes are observed near the typhoon region. Close to the cyclone eye wind speed data (Figure A5**e**) indicate strong winds up to 18 m/s. Comparing past and present analogues of Typhoon Hinnamnor, we do not find significant changes in pressure (Figure A5**f**), although larger geopotential heights (Figure A5**g**) up to 24 m are observed. We also observe increased precipitations up to 18 mm/day (Figure A5**h**), higher temperatures up to 2-2.5 °C (Figure A5**i**), and stronger winds, close to the cyclone center, up to 3 m/s (Figure A5**j**). Beyond these changes, we detect the likely influence of the AMO natural tele-connection pattern (Figure A6**b**). Therefore, as with Storm Poly, the contribution of internal climate variability cannot be neglected. Meanwhile we do not detect any significant changes in their seasonal occurrence (Figure A6**d**). Despite the

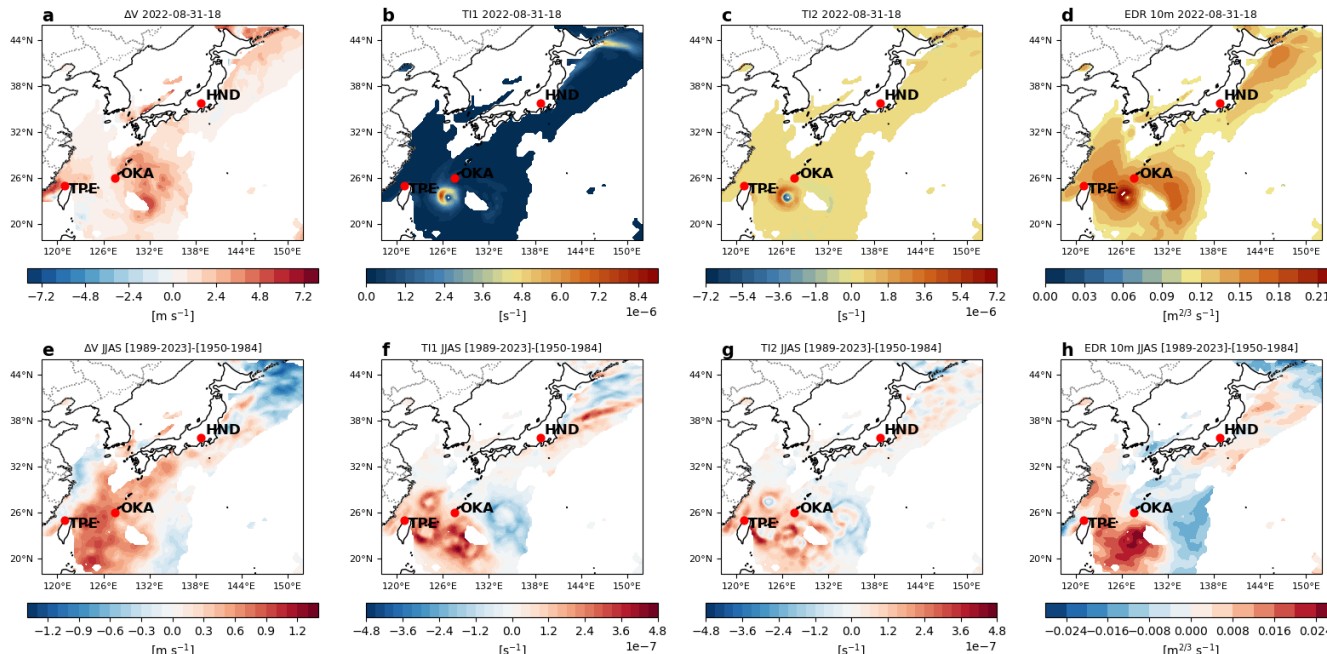

**Figure 7.** Analogue-based results on aviation-related metrics for the Typhoon Hinnamnor. Vertical wind speed difference **a**, Ellrod index TI1 **b**, Ellrod index TI2 **c**, and EDR **d** at the cyclone time. Difference between factual [1989-2023] and counterfactual [1950-1984] period of the average composites at the time-steps corresponding to the analogues for vertical wind speed difference **e**, Ellrod index TI1 **f**, Ellrod index TI2 **g**, and EDR **h**. White areas indicate regions with V at the cyclone time lower than 7 m/s. Red dots indicate major airports in the region: Taipei-Taiwan Taoyuan International Airport (TPE), Naha International Airport (OKA) and Haneda Airport (HND).

unique characteristics of each event in terms of meteorological variables, a common trend emerges when inspecting aviation-related impact variables. Specifically, turbulence-related metrics (TI1, TI2) (see Figures 5-7) show a significant increase over a wide area within the selected domain during the present period. Moreover, this intensification is often associated with higher EDR values and stronger wind shear. In details, the storm Poly generated strong (up to 14 m/s) vertical wind speed gradients (Figure 5**a**) in a region further south than the center of the cyclone, nearby Amsterdam. Over the same regions large values





of all turbulence-related metrics, i.e., TI1, TI2, and EDR, are observed (Figure 5**b-d**). By comparing present episodes with past events we observe increased velocity shears up to 1 m/s manifested as a horizontal belt stretching from the west coast of the UK to northern Poland (Figure 5**e**), likely affecting huge airports as Amsterdam and Copenhagen. Over the same regions but with different localized effects, larger increases are found in TI1, TI2, and EDR (Figure 5**f-h**). Specifically, TI1 is larger close to Copenhagen airport, thus suggesting stronger wind shears far away from the cyclone eye; conversely, TI2 increases

close to Amsterdam airport, thus highlighting a role of the frontogenesis due to the cyclone center. For the 2023 February North American windstorm (Figure 6) we do not depict significant increases in turbulence-related metrics. Finally, for the Typhoon Hinnamnor (Figure 7) we observe strongest winds impacted the northeastern coasts of Japan and the East China Sea, particularly near the eastern coast of Taiwan. Larger wind speed gradients (up to +1.5 m/s), along with higher TI1, TI2 and EDR values, are found in the same region, thus highlighting more turbulent scenarios in the present climate compared to past

conditions.

## 4 Discussion and conclusions

This study performs an analysis of four recent storms – Eunice, Poly, a North American windstorm, and Typhoon Hinnamnor – that occurred over Europe, the USA, and East Asia between 2022 and 2023, with a focus on Storm Eunice. We show how storms likely causing flight disruptions are becoming more intense, with analogues in the present period exhibiting stronger

ground winds and wind shears, thereby increasing risks for the aviation sector.

The analogue-based analysis of Storm Eunice provides compelling evidence of how extratropical cyclones impacting Europe are evolving under current climate conditions. Our results indicate that Eunice-like storms are not only more intense in the present climate but also more frequent during the early winter season, shifting towards December. This finding aligns with previous studies (e.g., Ginesta et al., 2024), reinforcing the notion that climate change is influencing the characteristics of

extreme storm events. The analysis of meteorological variables reveals a systematic increase in geopotential height anomalies and surface-level pressure patterns, suggesting a north-eastward shift in storm trajectories compared to past analogues. This shift, in turn, influences precipitation distribution, leading to enhanced rainfall over Ireland and parts of the UK. These changes indicate a potential increase in flood risk and infrastructural stress due to more intense and frequent storm activity. Our assessment of wind speed anomalies further supports the hypothesis that present-day storms exhibit stronger wind fields, particularly

over key aviation hubs such as Heathrow and Dublin airports. The increase in wind speeds, combined with enhanced vertical wind shear, presents substantial challenges for aviation operations. The analysis of turbulence indices (TI1, TI2, and EDR) highlights increased turbulence intensity over the southern UK and northern France, particularly near coastal regions. These factors contribute to heightened risks for aircraft during take-off and landing, necessitating improved forecasting tools and operational strategies to mitigate the impacts on air traffic. One of the most significant implications of our findings is the po-

tential influence of climate change on the intensification of extreme weather events. The fact that no significant changes were observed in ENSO, AMO, or PDO distributions for the analogues suggests that the detected variations are primarily driven by

anthropogenic climate change rather than natural climate variability. This conclusion underscores the urgency of incorporating evolving storm characteristics into risk assessment frameworks for transportation infrastructure, particularly for aviation safety.

Future research should focus on refining methodologies for storm attribution by integrating high-resolution models that better capture the mesoscale processes involved in storm evolution. Additionally, further studies should investigate the impact of these changing storm patterns on other sectors, such as energy supply, emergency response, and insurance risk assessment. A broader multi-disciplinary approach will be essential for developing adaptive strategies that enhance resilience against increasingly extreme weather events.

    In summary, our findings demonstrate that Eunice-like storms are becoming more severe in the current climate, with signifi-
cant implications for transportation systems and public safety. By enhancing predictive capabilities and implementing adaptive measures, stakeholders can mitigate the risks associated with these evolving meteorological threats.

*Code availability.*    All codes used for the analysis and generating the figures can be obtained from the authors upon request.

*Data availability.*    The data that support the findings of this study are openly available. ERA5 is the latest climate reanalysis being produced by ECMWF as part of implementing the EU- funded Copernicus Climate Change Service (C3S), providing hourly data on atmospheric,
land-surface and sea-state parameters together with estimates of uncertainty from 1979 to present day. ERA5 data are available on the C3S Climate Data Store on regular latitude-longitude grids at 0.25° x 0.25° resolution at https://cds.climate.copernicus.eu/#!/home.

*Author contributions.*    LR performed the analyses, analyzed the results and wrote the draft of the manuscript. TA and DF conceived the study. All authors contributed to discuss the results and to write the paper.

*Competing interests.*    The authors declare that they have no conflict of interest.

*Acknowledgements.*    TA and DF acknowledge useful discussions within the MedCyclones COST Action (CA19109) and the FutureMed COST Action (CA22162) communities. DF acknowledge the European Union Horizon 2020 research and innovation programme under grant agreement no. 101003469 (XAIDA).

**Appendix A: Supplementary Figures**




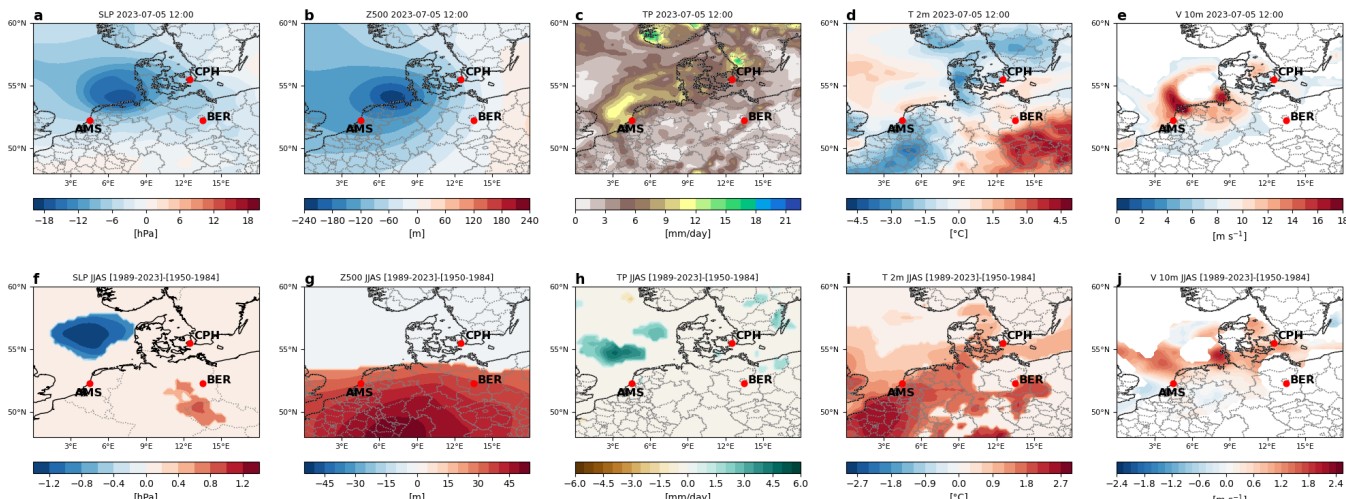

**Figure A1.** Analogue-based results for the Storm Poly. SLP **a** anomaly, Z500 **b** anomaly, TP **c**, 2-m T **d** anomaly and 10-m wind speed **e** at the cyclone time. Difference between factual [1989-2023] and counterfactual [1950-1984] period of the average anomalies at the time-steps corresponding to the analogues for SLP **f**, Z500 **g**, TP **h**, 2-m temperature **i** and 10-m wind speed **j**. In the second row, shadings indicate significant changes. Red dots indicate major airports in the region: Amsterdam Airport Shipol (AMS), Copenhagen Airport (CPH) and Berlin Brandenburg Airport (BER).

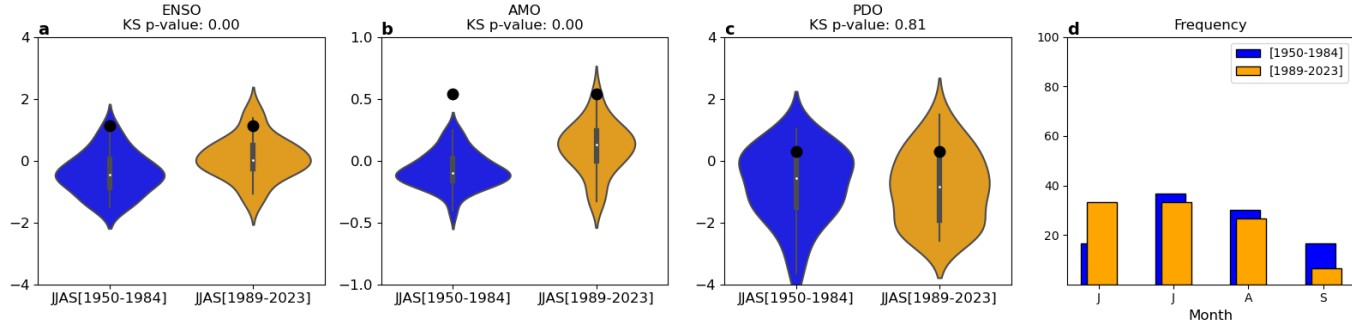

**Figure A2.** Analogue-based results for the Storm Poly. Violin plots showing the distribution of monthly index values for ENSO **a**, AMO **b**, and PDO **c** during the past (blue) and present (yellow) periods. Values for the peak day of the extreme event are marked by a black dot. Panel **d**: distribution of the frequency (%) of analogues occurring in each sub-period.

## References

Alberti, T., , Faranda, D., Rapella, L., Coppola, E., Lepreti, F., Dubrulle, B., and Carbone, V.: Impacts of Changing Atmospheric Circulation Patterns on Aviation Turbulence Over Europe, Geophysical Research Letters, 51, 10475, https://doi.org/10.1038/s41598-023-36816-8, 2024.



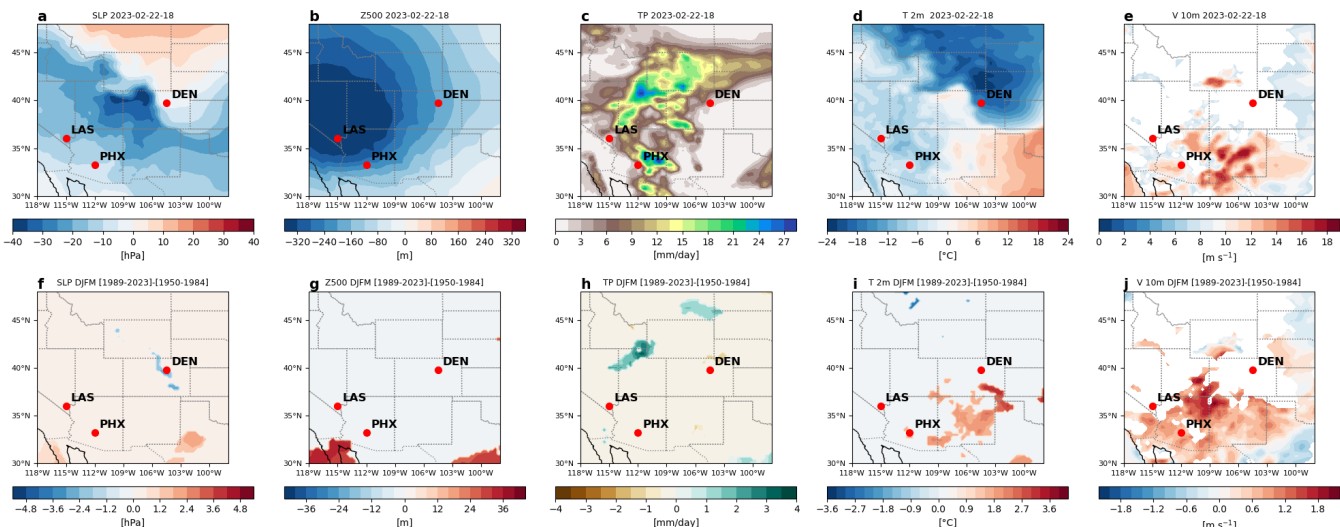

**Figure A3.** Analogue-based results for the February 2023 USA windstorm. SLP **a** anomaly, Z500 **b** anomaly, TP **c**, 2-m T **d** anomaly and 10-m wind speed **e** at the cyclone time. Difference between factual [1989-2023] and counterfactual [1950-1984] period of the average anomalies at the time-steps corresponding to the analogues for SLP **f**, Z500 **g**, TP **h**, 2-m temperature **i** and 10-m wind speed **h**. In the second row, shadings indicate significant changes. Red dots indicate major airports in the region: McCarran International Airport (LAS), Phoenix Sky Harbor International Airport (PHX) and Denver International Airport (DEN).

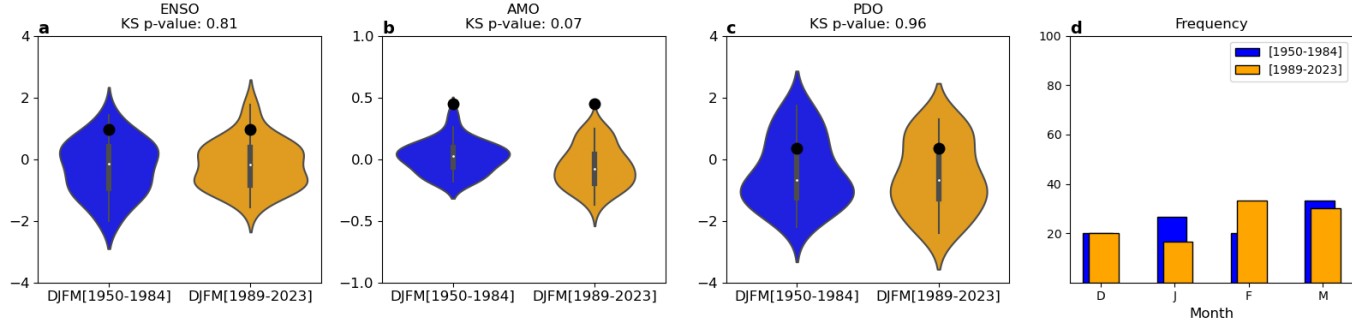

**Figure A4.** Analogue-based results for the February 2023 USA windstorm. Violin plots showing the distribution of monthly index values for ENSO **a**, AMO **b**, and PDO **c** during the past (blue) and present (yellow) periods. Values for the peak day of the extreme event are marked by a black dot. Panel **d**: distribution of the frequency (%) of analogues occurring in each sub-period.

Archer, C. L. and Caldeira, K.: Historical trends in the jet streams, Geophysical Research Letters, 35, 2008.

Borsky, S. and Unterberger, C.: Bad weather and flight delays: The impact of sudden and slow onset weather events, Economics of trans-

portation, 18, 10–26, 2019.

Burbidge, R.: Climate change risks and resilience for European aviation, Transportation Research Procedia, 72, 3276–3282, 2023.

Changnon, S. A.: Effects of summer precipitation on urban transportation, Climatic Change, 32, 481–494, 1996.



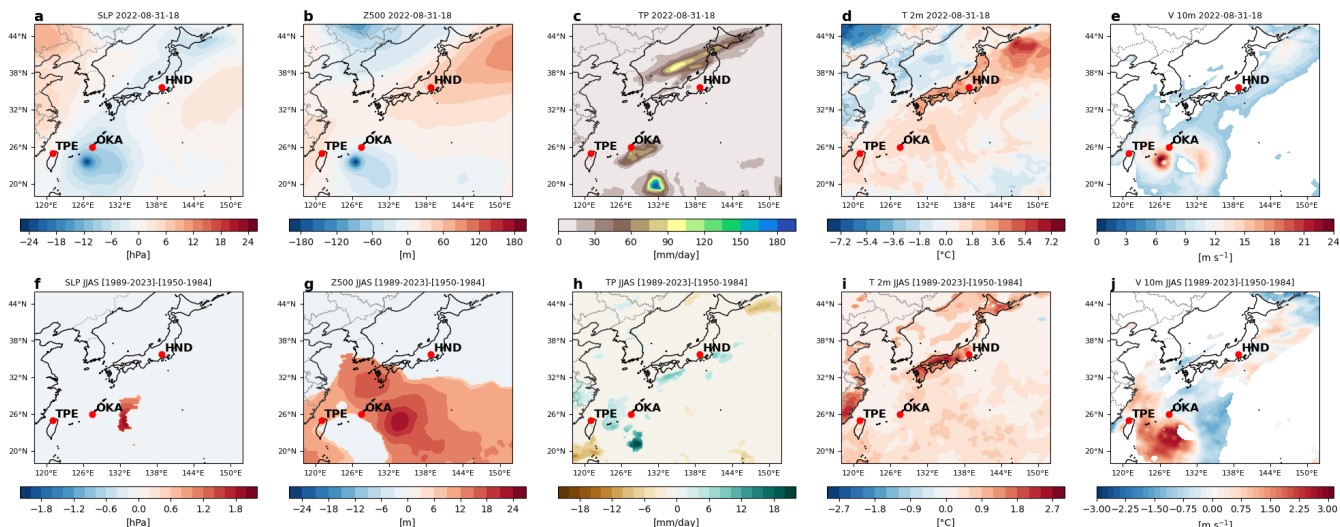

**Figure A5.** Analogue-based results for the Typhoon Hinnamnor. SLP **a** anomaly, Z500 **b** anomaly, TP **c**, 2-m T **d** anomaly and 10-m wind speed **e** at the cyclone time. Difference between factual [1989-2023] and counterfactual [1950-1984] period of the average anomalies at the time-steps corresponding to the analogues for SLP **f**, Z500 **g**, TP **h**, 2-m temperature **i** and 10-m wind speed **j**. In the second row, shadings indicate significant changes. Red dots indicate major airports in the region: Taipei-Taiwan Taoyuan International Airport (TPE), Naha International Airport (OKA) and Haneda Airport (HND).

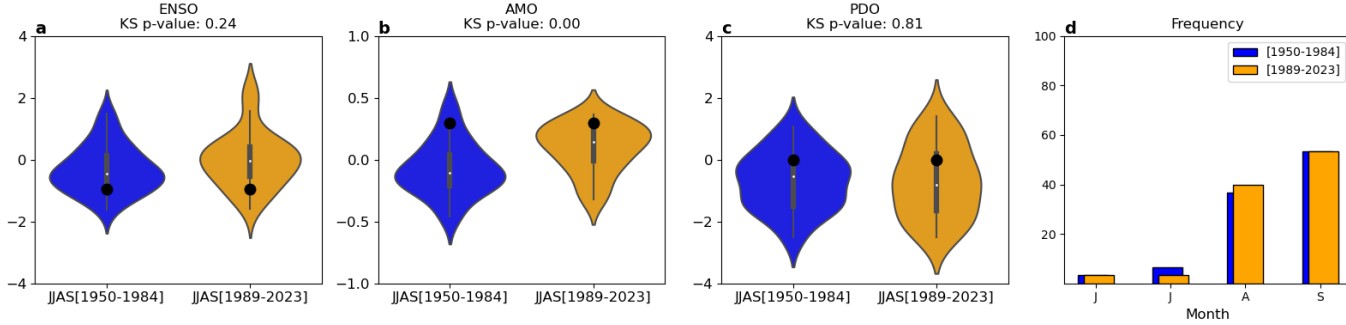

**Figure A6.** Analogue-based results for the Typhoon Hinnamnor. Violin plots showing the distribution of monthly index values for ENSO **a**, AMO **b**, and PDO **c** during the past (blue) and present (yellow) periods. Values for the peak day of the extreme event are marked by a black dot. Panel **d**: distribution of the frequency (%) of analogues occurring in each sub-period.

Cheung, H. M. and Chu, J.-E.: Global increase in destructive potential of extratropical transition events in response to greenhouse warming, npj Climate and Atmospheric Science, 6, 137, 2023.

Cornman, L. B. and Carmichael, B.: Varied research efforts are under way to find means of avoiding air turbulence., ICAO Journal, 1993.

Ellrod, G. P. and Knapp, D. I.: An Objective Clear-Air Turbulence Forecasting Technique: Verification and Operational Use, Weather and Forecasting, 7, 150–165, https://doi.org/10.1175/1520-0434(1992)007<0150:AOCATF>2.0.CO;2, 1992.



Ellrod, G. P. and Knox, J. A.: Improvements to an Operational Clear-Air Turbulence Diagnostic Index by Addition of a Divergence Trend Term, Weather and Forecasting, 25, 789–798, https://doi.org/10.1175/2009WAF2222290.1, 2010.

Faranda, D., Ginesta, M., Alberti, T., Coppola, E., and Anzidei, M.: Attributing Venice Acqua Alta events to a changing climate and evaluating the efficacy of MoSE adaptation strategy, npj Climate and Atmospheric Science, 6, 181, https://doi.org/10.1038/s41612-023-00513-0, 2023.

Faranda, D., Messori, G., Coppola, E., Alberti, T., Vrac, M., Pons, F., Yiou, P., Saint Lu, M., Hisi, A. N., Brockmann, P., et al.: ClimaMeter: contextualizing extreme weather in a changing climate, Weather and Climate Dynamics, 5, 959–983, 2024.

Federal Aviation Administration: Advisory Circular AC 00-24C: Thunderstorms, Federal Aviation Administration, available online: https://www.faa.gov/documentlibrary/media/advisory_circular/ac%2000-24c.pdf [Accessed November 26, 2024], 2022.

Fery, L. and Faranda, D.: Analysing 23 years of warm-season derechos in France: a climatology and investigation of synoptic and environmental changes, Weather and Climate Dynamics, 5, 439–461, 2024.

Forbis, D. C., Patricola, C. M., Bercos-Hickey, E., and Gallus Jr, W. A.: Mid-century climate change impacts on tornado-producing tropical
cyclones, Weather and Climate Extremes, 44, 100 684, 2024.

Ginesta, M., Flaounas, E., Yiou, P., and Faranda, D.: Anthropogenic climate change will intensify European explosive storms analogous to Alex, Eunice, and Xynthia, Journal of Climate, 37, 5427–5452, 2024.

Gratton, G. B., Williams, P. D., Padhra, A., and Rapsomanikis, S.: Reviewing the impacts of climate change on air transport operations, The Aeronautical Journal, 126, 209–221, 2022.

Gulev, S. K., Thorne, P. W., Ahn, J., Dentener, F. J., Domingues, C. M., Gerland, S., Gong, D., Kaufman, D. S., Nnamchi, H. C., Quaas, J., et al.: Changing state of the climate system, 2021.

Hersbach, H., Bell, B., Berrisford, P., Biavati, G., Horányi, A., Muñoz Sabater, J., Nicolas, J., Peubey, C., Radu, R., Rozum, I., Schepers, D., Simmons, A., Soci, C., Dee, D., and Thépaut, J.-N.: ERA5 monthly averaged data on single levels from 1940 to present, [Dataset], https://doi.org/10.24381/cds.f17050d7, (Accessed on 09-11-2022), 2023.

Hsiao, C.-Y. and Hansen, M.: Econometric analysis of US airline flight delays with time-of-day effects, transportation research Record, 1951, 104–112, 2006.

International Civil Aviation Organization: Safety Aspects of Tailwind Operations, Tech. rep., ICAO, https://skybrary.aero/sites/default/files/bookshelf/1148.pdf, accessed: 2025-02-20, 2008.

International Civil Aviation Organization: Seventh Meeting of the MOG, https://www.icao.int/airnavigation/METP/Seventh%
20Meeting%20of%20the%20MOG/METPWGMOG-7_Attachment_SN_35_METP_4_WP.xxxx.Revised_EDR_values_Amd_79_vs_1.0_22mar2018.docx, accessed: 2024-07-12, 2018.

Kendon, M.: Storms Dudley, Eunice and Franklin February 2022, Tech. rep., Technical Report. Met Office, 2022.

Kim, K., Lee, H., Lee, M., Bae, Y. H., Kim, H. S., and Kim, S.: Analysis of weather factors on aircraft cancellation using a multilayer complex network, Entropy, 25, 1209, 2023.

Kim, Y.-H., Lee, M., Min, S.-K., Park, D.-S. R., Cha, D.-H., Byun, Y.-H., and Heo, J.: Global Warming–Induced Warmer Surface Water over the East China Sea Can Intensify Typhoons like Hinnamnor, Bulletin of the American Meteorological Society, 105, E1416 – E1421, https://doi.org/10.1175/BAMS-D-23-0240.1, 2024.

Lee, J.-Y., Marotzke, J., Bala, G., Cao, L., Corti, S., Dunne, J. P., Engelbrecht, F., Fischer, E., Fyfe, J. C., Jones, C., et al.: Future global climate: scenario-based projections and near-term information, in: Climate change 2021: The physical science basis. Contribution of





working group I to the sixth assessment report of the intergovernmental panel on climate change, pp. 553–672, Cambridge University Press, 2021.

Luu, L. N., Vautard, R., Yiou, P., van Oldenborgh, G. J., and Lenderink, G.: Attribution of extreme rainfall events in the south of France using EURO-CORDEX simulations, Geophysical Research Letters, 45, 6242–6250, 2018.

Marvel, K. and Bonfils, C.: Identifying external influences on global precipitation, Proceedings of the National Academy of Sciences, 110, 360     19 301–19 306, 2013.

Masson-Delmotte, V., Zhai, P., Pirani, A., Connors, S. L., Péan, C., Berger, S., Caud, N., Chen, Y., Goldfarb, L., Gomis, M., et al.: Climate Change 2021: The Physical Science Basis. Contribution of Working Group I to the Sixth Assessment Report of the Intergovernmental Panel on Climate Change, vol. 2, Cambridge University Press, Cambridge, United Kingdom and New York, NY, USA, https://doi.org/10.1017/9781009157896, 2021.

Oo, K. and Oo, K.: Analysis of the most common aviation weather hazard and its key mechanisms over the Yangon flight information region, Advances in Meteorology, 2022, 5356 563, 2022.

Otto, F.: Attribution of extreme weather events: how does climate change affect weather?, Weather, 74, 325–326, 2019.

Otto, F. E. L.: Extreme events: The art of attribution, Nat. Clim. Chang., 6, 342–343, 2016.

Priestley, M. D. and Catto, J. L.: Improved representation of extratropical cyclone structure in HighResMIP models, Geophysical Research 370     Letters, 49, e2021GL096 708, 2022.

Robinson, P. J.: The influence of weather on flight operations at the Atlanta Hartsfield International Airport, Weather and forecasting, 4, 461–468, 1989.

Rosanes, M.: Storms Dudley and Eunice: insured loss estimates revealed, Insurance Business, https://www.insurancebusinessmag.com/uk/ news/catastrophe/storms-dudley-and-eunice-insured-loss-estimates-revealed-326851.aspx, accessed: 2025-01-22, 2022.

Sasse, M. and Hauf, T.: A study of thunderstorm-induced delays at Frankfurt Airport, Germany, Meteorological Applications, 10, 21–30, 2003.

Sharman, R. and Lane, T.: Aviation turbulence, Springer International Publishing, Switzerland, DOI, 10, 978–3, 2016.

Storer, L. N., Williams, P. D., and Joshi, M. M.: Global response of clear-air turbulence to climate change, Geophysical Research Letters, 44, 9976–9984, 2017.

Storer, L. N., Williams, P. D., and Gill, P. G.: Aviation turbulence: dynamics, forecasting, and response to climate change, Pure and Applied Geophysics, 176, 2081–2095, 2019.

The Sun: Storm Eunice Ireland – Flights cancelled and roads closed as red alert issued, https://www.thesun.ie/news/8384038/ storm-eunice-ireland-flights-cancelled-roads/, accessed: 2025-02-20, 2022.

Timmins, B.: Storm Eunice: Flights and train services cancelled, BBC, https://www.bbc.com/news/business-60430197, accessed: 2025-01- 385     22, 2022.

Ulbrich, U., Leckebusch, G. C., and Pinto, J. G.: Extra-tropical cyclones in the present and future climate: a review, Theoretical and applied climatology, 96, 117–131, 2009.

Vautard, R., Christidis, N., Ciavarella, A., Alvarez-Castro, C., Bellprat, O., Christiansen, B., Colfescu, I., Cowan, T., Doblas-Reyes, F., Eden, J., et al.: Evaluation of the HadGEM3-A simulations in view of detection and attribution of human influence on extreme events in Europe, 390     Climate Dynamics, 52, 1187–1210, 2019.

Volonté, A., Gray, S. L., Clark, P. A., Martínez-Alvarado, O., and Ackerley, D.: Strong surface winds in Storm Eunice. Part 1: storm overview and indications of sting jet activity from observations and model data, Weather, 79, 40–45, 2024a.





Volonté, A., Gray, S. L., Clark, P. A., Martínez-Alvarado, O., and Ackerley, D.: Strong surface winds in Storm Eunice. Part 2: airstream analysis, Weather, 79, 54–59, 2024b.

Williams, J. K.: Using random forests to diagnose aviation turbulence, Machine learning, 95, 51–70, 2014.

Williams, P. D.: Increased light, moderate, and severe clear-air turbulence in response to climate change, Advances in Atmospheric Sciences, 34, 576–586, 2017.

Williams, P. D. and Joshi, M. M.: Intensification of winter transatlantic aviation turbulence in response to climate change, Nature Climate Change, 3, 644–648, https://doi.org/10.1038/nclimate1866, 2013.

Wu, L., Zhao, H., Wang, C., Cao, J., and Liang, J.: Understanding of the effect of climate change on tropical cyclone intensity: A Review, Advances in Atmospheric Sciences, 39, 205–221, 2022.

Yiou, P., Jézéquel, A., Naveau, P., Otto, F. E. L., Vautard, R., and Vrac, M.: A statistical framework for conditional extreme event attribution, Advances in Statistical Climatology, Meteorology and Oceanography, 3, 17–31, https://doi.org/10.5194/ascmo-3-17-2017, 2017.

Zappa, G., Shaffrey, L. C., Hodges, K. I., Sansom, P. G., and Stephenson, D. B.: A multimodel assessment of future projections of North
Atlantic and European extratropical cyclones in the CMIP5 climate models, Journal of Climate, 26, 5846–5862, 2013.