# Peer review of "Anthropogenic climate change has increased severity of mid-latitude storms and impacted airport operations"

_EGUsphere, 2025_

## Author Comment (AC1)

**RESPONSE TO REVIEWER RC1**

May 26, 2025

Dear Editor,
we thank the Reviewer and the Editor for the time dedicated to reading and revising our manuscript. We have taken great care of answering your comments. Among all modifications, we stress the main ones here. We provide our full answer letter below.

Sincerely,
Lia Rapella,
on behalf of the authors

**General comments**

**Comment #1**

The link between the impacts of the effects of climate change and airport operations are superficial and sometimes misleading (e.g., stronger headwind actually improves airplane performance; the authors also suggest an increased risk due to increased tailwinds, which is not likely a big impact because tailwind operations are easily and commonly avoided); while wind shear in downbursts is dangerous, in extratropical cyclones wind shear typically means that the planes take off into the shear, which improves climb performance (likewise, descending into wind shear acts to increase the headwind, so it is not inherently dangerous). So, a much more nuanced analysis is needed when considering the impacts of ETCs on aircraft operations. (I'm not arguing that stronger winds don't cause disruptions, but the current argument in the manuscript is imprecise.)
My recommendation is that either the "spin" of the paper is adjusted, away from airport operations, and instead putting more emphasis on how the weather systems were affected by climate change. Alternatively, a much more detailed analysis about how climate change actually affected air travel would be needed: Given the increases of turbulence/wind speed, how many more delays happened due to climate change? What are the costs? Wouldn't this be the questions that stakeholders would be interested in?.

**We agree that the target of our paper is not directly searching impacts of climate change on airports' operation, thus we will correspondingly adjust the target and writing of the paper, as suggested by**

the Reviewer, in the revised version. We would like to emphasize that our analysis is indeed an obervation-based (analogues) attribution approach to investigate how the weather systems that contributed and forced to airport closures, disruptions, delays, and cancellations, have changed compared to similar synoptic-scale situations and conditions occurred in the past. This means that, although similar large-scale patterns are observed historically, their effects and impacts, as measured, among others, by changing wind patterns and intensity, wind shear and bulk wind difference, turbulence-related metrics, are now different with respect to the past, thus suggesting a possible link with anthropogenic climate change. For this reason, we focus on 4 high-impact weather events, geographically distributed over Europe, USA, and Asia, that forced airports' disruptions, to find if these impacts belongs to the same class of previously-observed ones. Indeed, a detailed analysis on impacts on air travel due to climate change would require to completely revise the focus of our paper, also adapting our methodology to changing conditions that cannot be ruled out in an objective way. For example, while we can quantify natural mode variability and/or climate change trends using indices and long-term behaviour of key meteorological parameters to introduce an attribution-based framework, widely employed into different sectors (e.g., Faranda et al., 2024), directly investigated impacts on air travel in terms of costs, delays, and weather disruptions would require to add more data, some of them not freely available or covering short time intervals to be statistically robust to assess differences among present and past conditions, as well as, to take (and model/parametrize) into account additional factors as the increased number of flights over time, significantly higher today than decades ago, the different types of aircraft employed now with respect to those used earlier. Thus, we favourably accept the suggestion of the Reviewer to adjust the wording and general organization of the paper to clearly highlight our aims.

**Comment #2**

Throughout the manuscript, the authors use wind shear (which should have units of 1/s rather than m/s) when I think they mean "bulk wind difference" (which has the unit of m/s).

We fully agree with the Reviewer and we will change accordingly.

**Specific comments**

We will implement corrections to each specific comment.

---

## Author Comment (AC2)

**RESPONSE TO REVIEWER RC2**

May 26, 2025

Dear Editor,
we thank both Reviewer and the Editor for the time dedicated to reading and revising our manuscript. We have taken great care of answering your comments. Among all modifications, we stress the main ones here. We provide our full answer letter below.

Sincerely,
Lia Rapella,
on behalf of the authors

**Main comments**

**Comment #1**

This first comment relates to the analogue method.
As I understand it, the analogue method only considers the time of maximum intensity when looking for similar cyclones in the earlier period. I wonder how different the analogues would be (and therefore also the results) if you took into account the cyclone development. In terms of impacts, particularly for precipitation, the development period of the cyclone may matter more in terms of overall similarity than only the time of maximum intensity. I assume it should not be too difficult to look for analogues at various times before maximum intensity to see if the results are sensitive to this. Finding analogues that remained similar throughout their lifecycle may find cyclones that are more representative of the cyclones from the current period.

**We particularly thank the Reviewer for raising this point. It has been shown (e.g., Ginesta et al., 2024) that attribution-based results on analogues are robust also if taking into account time steps related to the cyclone development and not just the time at which the minimum in SLP is observed. However, in the revised version we will assess this point by looking also for analogues at times before and after the defined cyclone time (see Figures 1)-4).**

[Figure]

Figure 1: Analogue-based results for the Storm Eunice on 2022-02-18 00:00 (before the cyclone time). SLP **a** anomaly, V 10m **b** and TP **c**. Difference between factual [1989-2023] and counterfactual [1950-1984] period of the average SLP anomalies **d**, V 10m **e** and TP **f**. Bulk wind difference **g**, TI1 **h**, TI2 **i** and difference between factual [1989-2023] and counterfactual [1950-1984] period of the same variables $\Delta V$ **j**, TI1 **k** and TI2 **l**. In the second and fourth rows, shadings indicate significant changes. Red dots indicate major airports in the region: Dublin Airport (DUB), Manchester Airport (MAN) and Heathrow Airport (LHR).

[Figure]

Figure 2: Analogue-based results for the Storm Eunice on 2022-02-18 12:00 (after the cyclone time). SLP **a** anomaly, V 10m **b** and TP **c**. Difference between factual [1989-2023] and counterfactual [1950-1984] period of the average SLP anomalies **d**, V 10m **e** and TP **f**. Bulk wind difference **g**, TI1 **h**, TI2 **i** and difference between factual [1989-2023] and counterfactual [1950-1984] period of the same variables $\Delta V$ **j**, TI1 **k** and TI2 **l**. In the second and fourth rows, shadings indicate significant changes. Red dots indicate major airports in the region: Dublin Airport (DUB), Manchester Airport (MAN), Heathrow Airport (LHR) and Amsterdam Airport (AMS).

[Figure]

Figure 3: Analogue-based results for the Storm Eunice on 2022-02-18 18:00 (after the cyclone time). SLP **a** anomaly, V 10m **b** and TP **c**. Difference between factual [1989-2023] and counterfactual [1950-1984] period of the average SLP anomalies **d**, V 10m **e** and TP **f**. Bulk wind difference **g**, TI1 **h**, TI2 **i** and difference between factual [1989-2023] and counterfactual [1950-1984] period of the same variables $\Delta V$ **j**, TI1 **k** and TI2 **l**. In the second and fourth rows, shadings indicate significant changes. Red dots indicate major airports in the region: Manchester Airport (MAN), Heathrow Airport (LHR), Amsterdam Airport (AMS) and Copenhagen Airport (CPH).

[Figure]

Figure 4: Analogue-based results for the Storm Eunice on 2022-02-19 00:00 (after the cyclone time). SLP **a** anomaly, V 10m **b** and TP **c**. Difference between factual [1989-2023] and counterfactual [1950-1984] period of the average SLP anomalies **d**, V 10m **e** and TP **f**. Bulk wind difference **g**, TI1 **h**, TI2 **i** and difference between factual [1989-2023] and counterfactual [1950-1984] period of the same variables $\Delta V$ **j**, TI1 **k** and TI2 **l**. In the second and fourth rows, shadings indicate significant changes. Red dots indicate major airports in the region: Amsterdam Airport (AMS), Copenhagen Airport (CPH) and Berlin Brandenburg Airport (BER).

[Figure]

[Figure]

[Figure]

Figure 5: Violin plots showing the distribution of daily index values for SCAN **a**, NAO **b**, and EA **c** during the past (blue) and present (yellow) periods. Values for the peak day of the extreme event are marked by a black dot.

**Comment #2**

This comment relates to the modes of variability considered.
When accounting for low frequency variability you consider ENSO, the AMO and PDO indices. I was surprised you did not include the NAO as this is well known to greatly impact both the frequency and intensity of extratropical cyclones (granted for the North Atlantic/Europe primarily). Have you tested whether including the NAO in your assessment of natural variability changes your results? Some discussion relating to this is needed regardless.

**We particularly thank the Reviewer for raising this point. In the revised version we will assess the low-frequency climate variability also including additional tele-connection indices as the suggested NAO, but also the East Atlantic (EA) and the Scandinavian (SCAN) pattern which are known to influence Euro-Mediterranean weather extremes. By performing statistical tests we note that for the case of Eunice (see Figure 5) only the EA pattern is significant, thus suggesting a very minor role of natural low-frequency variability in this event (one mode over six).**

**Minor comments**

**We will implement all corrections related to each minor/technical comment.**

---

## Referee Report (RR1)

**Overview**

The authors present an interesting study on the intensification of various extreme weather events from recent years. The analysis is interesting, although I would say not especially novel or ground-breaking considering previous work in this area particularly from the group of one of the authors. The novel aspect is the impact on airport operations, which is interesting, however I do question the direct role or systematic impact of these cyclones in recent years. A lot of the impact of the cyclones is down to the location of their track and this is unlikely to be a systematic increase either in the recent historical period or as the climate continues to warm. Therefore, I question if the airport impact and perceived increase is an actual increase or more just internal/natural variability? This is my only major comments that I would like the authors to address and discuss in this study. My other minor comments are detailed below.

- 1. L38 some references to support these statements would be good
- 2. L75/76 are these events that you have chosen actually high impact events representative of the current climate, or are they big outliers of the present day distribution? It would be good to illustrate how the actual events compare to the analogues that you are showing. Are they more intense/weaker? How does the wind speeds compare?
- 3. Table 1 can you be more quantitative with the values in this table. Simply stating "several" or "few" flights diverted is not especially evidential.
- 4. L93 why did you choose not to go back to 1940 with ERA5 for your analysis? Surely this would help you find more analogues and prove your case even further?
- 5. L105 what is this proxy? You need to introduce this metric
- 6. L189-190 it appears in fig. 2 that the strongest winds are weaker in the present day storms could the authors also comment on this please as it somewhat disagrees with the "higher intensity" message trying to be communicated
- 7. L199 the more "extensive regions" of strong winds is in agreement with findings of windstorm footprints being larger in a warmer climate (see recent papers by Dolores-Tesillos et al., 2022 (https://doi.org/10.5194/wcd-3-429-2022) and Priestley et al., 2024 (https://doi.org/10.1002/qj.4849)) and should be discussed.
- 8. L219-220 there is some disagreement as the SLP centre appears deeper in Figure A2. Please discuss this and ensure that the stages of the timeline that you say are consistent are actually consistent.
- 9. L246 storm Poly occurs in the Summer and yet here you discuss the winter. Are the analogues you have created for a different season as if so I would say this is not comparable. In Figure S6 you have JJAS, so please make sure there is a consistency (I imagine this is just a mis-naming in the main text) and that this is resolved
- 10. L260/261 As the above comment, please check the seasons and months being quoted as I again assume you mean DJFM?
- 11. L277/278 this widespread increase is not as apparent (or not that strong) in Figure 6 so please make this clear
- 12. L303/304 you state the SLP is shallower in the text so this is not consistent with the increased intensity you quote here. Please be specific about what features you are referring to.
- 13. L322-324 This argument I have difficulty with. There is still internal variability in the different states of the large-scale modes that could be causing this variation. The risk

in the same NAO state can vary quite substantially. A good focus of this work needs to be on the fact that there is lots of internal variability, and even though these storms are getting more intense, the changes in track or location are likely to be as much of a contributor to change in risk as global warming.

---

## Author Response (AR2)

**RESPONSE TO REVIEWERS**

**August 5, 2025**

Dear Editor,

we thank the Reviewers and the Editor for the time dedicated to reading and revising our manuscript. We have taken great care of answering your comments. Corrections to the technical comments of Reviewers RC1 and RC2 can be found in the tracked manuscript.

For Reviewer RC3, we provide our full answer letter below.

Sincerely, Lia Rapella, on behalf of the authors

**General comments**

The authors present an interesting study on the intensification of various extreme weather events from recent years. The analysis is interesting, although I would say not especially novel or ground-breaking considering previous work in this area particularly from the group of one of the authors. The novel aspect is the impact on airport operations, which is interesting, however I do question the direct role or systematic impact of these cyclones in recent years. A lot of the impact of the cyclones is down to the location of their track and this is unlikely to be a systematic increase either in the recent historical period or as the climate continues to warm. Therefore, I question if the airport impact and perceived increase is an actual increase or more just internal/natural variability? This is my only major comments that I would like the authors to address and discuss in this study.

We thank the Reviewer for their thoughtful feedback and for recognizing the interest in our study, particularly the novel aspect regarding impacts on airport operations. We appreciate the point regarding the potential influence of cyclone track variability and the challenges of attributing observed impacts to systematic changes rather than internal variability. We have added more details in the revised manuscript to highlight the role of cyclone track variability and internal climate variability in modulating the impact of extreme weather

on specific locations, such as airports. While our results suggest an increasing trend in wind speed and turbulence due to extreme events at specific airports, which is mostly due to anthropogenic climate change (at least in a statistical sense), we agree that further attribution studies should focus on the phases of internal natural variability to deeply assess its role. At the present, we can assess that in our 4 events natural variability played a minor role, although each phase of the large-scale modes (e.g., NAO) can play a role in modulating storm behaviour and associated risks.

**Minor comments**

**0.1**

L38 – some references to support these statements would be good. We added some references, including the IPCC AR6 report (Chapter 11), to support.

**0.2**

L75/76 – are these events that you have chosen actually high impact events representative of the current climate, or are they big outliers of the present day distribution? It would be good to illustrate how the actual events compare to the analogues that you are showing. Are they more intense/weaker? How does the wind speeds compare?

We selected these events as they had significant impacts on airport operations with flights' delays and cancellations. However, we also assessed the differences between the 10-m wind patterns for each event with the composite of the analogues in the factual period (1989-2023). We can see clearly that each event is more intense in terms of wind speed close to the airports, thus strengthening the choice of the four events as representatives of extreme events with high impact on airport operations within the current climate. We modified the text to add this point.

**0.3**

Table 1 – can you be more quantitative with the values in this table. Simply stating "several" or "few" flights diverted is not especially evidential.

We removed the column "Flights diverted" in the table as there is no agreement nor accuracy in the sources about the number of diverted flights.

Figure 1: Difference between V 10m at the cyclone time and V 10m averaged over the time-steps corresponding to the analogues for the factual [1989-2023] period, for the 4 events analysed in the study.

**0.4**

L93 – why did you choose not to go back to 1940 with ERA5 for your analysis? Surely this would help you find more analogues and prove your case even further? This is because ERA5 data are reliable starting from 1950.

**0.5**

L105 – what is this proxy? You need to introduce this metric.

We introduced the metric directly and we formulated differently the sentence.

**0.6**

L189-190 – it appears in fig. 2 that the strongest winds are weaker in the present day storms – could the authors also comment on this please as it somewhat disagrees with the "higher intensity" message trying to be communicated.

We agree that looking at the cyclone eye the wind speed is lower but considering that there is a change in the track of the cyclone we can see spreader effects of the wind speed (as expected). Indeed, our results suggest that wind speed variations are broader over the domain, especially in the southern part.

**0.7**

L199 – the more "extensive regions" of strong winds is in agreement with findings of windstorm footprints being larger in a warmer climate (see recent papers by Dolores-Tesillos et al., 2022 (https://doi.org/10.5194/wcd-3-429-2022) and Priestley et al., 2024 (https://doi.org/10.1002/qj.4849)) and should be discussed.

We thank the reviewer for this suggestion. We have incorporated this evidence in the revised manuscript. Specifically, we now discuss the agreement of our findings with the literature, highlighting the expansion of windstorm footprints in a warmer climate, as documented by Priestley et al., 2024 and Dolores-Tesillos et al., 2022.

**0.8**

L219-220 – there is some disagreement as the SLP centre appears deeper in Figure A2. Please discuss this and ensure that the stages of the timeline that you say are actually consistent.

We thank the reviewer for the observation. We would like to clarify that the cyclone time selected for the analysis of Storm Eunice corresponds to the time-step when the SLP reached a minimum over the British Islands region. In fact, as stated in LL 172-73, "As our focus is on impacts on airports disruptions, we have selected the British Islands as the spatial domain for the attribution analysis, given the

significant disruptions experienced in this region". However, we acknowledge that that lower SLP values may have occurred over other regions during the development of the cyclone.

**0.9**

L246 – storm Poly occurs in the Summer and yet here you discuss the winter. Are the analogues you have created for a different season as if so I would say this is not comparable. In Figure S6 you have JJAS, so please make sure there is a consistency (I imagine this is just a mis-naming in the main text) and that this is resolved.

We thank the reviewer for the observation. We mis-named in the main text, the considered analogues for Storm Poly are in the summer season, JJAS. We implemented the corrections accordingly.

**0.10**

L260/261 – As the above comment, please check the seasons and months being quoted as I again assume you mean DJFM?

As the above response, we wrote the wrong names of the months. We corrected figure A8 substituting JJAS with DJFM.

**0.11**

L277/278 – this widespread increase is not as apparent (or not that strong) in Figure 6 so please make this clear.

We changed the text to make it clearer.

**0.12**

L303/304 – you state the SLP is shallower in the text so this is not consistent with the increased intensity you quote here. Please be specific about what features you are referring to.

We clarified we refer to wind speed and turbulence in the text.

**0.13**

L322-324 - This argument I have difficulty with. There is still internal variability in the different states of the large-scale modes that could be causing this variation. The risk in the same NAO state can vary quite substantially. A good focus of this work needs to be on the fact that there is lots of internal variability, and even though these storms are getting more intense, the changes in track or location are likely to be as much of a contributor to change in risk as global warming.

We acknowledge that internal variability within each phase of the large-scale modes (e.g., NAO) can play a role in modulating storm

behaviour and associated risks. While our results suggest that anthropogenic climate change is the dominant driver behind the observed changes, we agree that variations in storm tracks or locations—even within the same phase of a given mode—can significantly influence risk. We clarified this point in the revised manuscript.

---

## Author Response (AR3)

**RESPONSE TO REVIEWERS**

**September 15, 2025**

Dear Editor,

we thank you for the time dedicated to provid another full editorial review of our manuscript. We have taken great care of answering your comments. We provide our full answer letter below.

Sincerely, Lia Rapella, on behalf of the authors

**General comments**

**0.1**

After the major concern of R3, I miss an appropriate discussion already in the introduction of other factors influencing variability in impacts, such as internal climate variability, in particular interannual and decadal variability (which also affect large-scale patterns, that affect ETC occurrence, e.g. Dorrington and Strommen 2020). In that sense the major concern of R3 is not sufficiently adressed, yet. I ask to discuss the relevant literature in the introduction already to provide a more balanced introduction, given the relatively short comparison periods for factual and counterfactual periods, and caveats in ERA5 stability across these periods.

We added a discussion in the introduction to other factors influencing variability of impacts, including the suggested references.

**0.2**

Also the potential role of an evolving observational network should be discussed at some places (Data / Methods, Discussion). The study should build trust in the long term stability and robustness of trends in variables considered here, as well as the potential effect of an evolving observational network on the results of the attribution framework.

We included a discussion on the suitability of the ERA5 product for long-term trends and attribution studies, as well as, also more details on possible effects related to the consistency of the reanalysis products before the satellite era.

**0.3**

As described at the moment, the study is not easily reproducible! What is missing is more background on which times / regions contribute to the analogues. I.e. is the geographical extent fixed, or in a moving frame? (if moving land-sea mask, orography would affect the composite), what are the timesteps? How many individual cases contribute to the analogue? Some of the information might be found in the methodological references. But the paper must be self-standing, and the results reproducible. A list of the time steps in the analogue composites is the least what should be provided along with some more technical information on the geographic extent of analogues.

We added two tables in the appendix with the analogues time-steps, for each event and for both factual and counterfactual period. Following also the suggestions the Editor provided in the *specific comments* section (see comment 0.9), we added more information in section 2.2.

**Minor comments**

**0.4**

119. The reference from 1993 seems quite old regarding the fact that technology to detect and avoid convection has improved. Consider referring to a newer study.

We added two more recent and general references.

**0.5**

128-31. Here you are citing studies which investigate specific years or episodes which were characterised by an increase of a specific weather phenomon which had impact on air traffic. In the light of R3's major comment you should mention the fact that interannual variability can explain differences in the number of incidents in these specific years. Perhaps this is a good place to introduce interannual variability.

We added a discussion in the introduction to other factors influencing variability of impacts, including the suggested references.

**0.6**

172. For Table 1 the abbreviations of major airports need to be introduced by first naming the cities or naming the cities in the table (caption) (as in Figure 1). In Figure 1 the introduction of abbreviations comes too late.

We have made explicit the airports abbreviations with the names of the cities in the caption of Table 1 and we reorganized the caption of Figure 1.

**0.7**

l110. As WCD is a journal with focus more on fundamental research rather than applied research I want to ask to actually show the formulae for TI1 and TI2, explaining with variables also which shear and deformation definition you mean and to explain how you compute it from ERA5 output. Also rather show formula as numbered equations than inline (also pertains to EDR 1 119).

We made explicit the formulas for TI1 and T12, showing them with numbered equations, as well as for EDR.

**0.8**

As you are using ERA5 for trend assessment of ERA5-derived impact variables, you have to discuss the long-term stability and how suitable ERA5 is for trend estimates in the variables used. Please discuss the relevant findings and supporting evidence from Bell et al. 2021, Soci et al. 2024 and, for the large-scale circulation, Simmons (2022). In particular the role of an evolving (surface) observational network for the detected trends in surface weather impacts must be discussed.

Did you use the latest release of the ERA5 back extension (Soci et al. 2024), which has improved track forecast for TCs in 1950s-1970s?

We added a new paragraph at the end of section 2.1:

"ERA5 provides a suitable basis for assessing trends in our impact variables, particularly when analyses are anchored in the well-observed satellite era and supported by basic robustness checks. Large-scale circulation trends since 1979 are physically consistent, dynamically plausible, and cross-supported by other reanalyses. As an example, Simmons et al. [4] showed strengthened and meridionally expanded tropical easterlies, shifts in the North Atlantic jet, and increases in extreme jet-stream winds, with patterns coherent across vertical levels and datasets. The recent back-extensions of ERA5 substantially increase temporal coverage, but their stability is lower because they rely on much sparser and evolving observations, especially before the satellite era ([1]; [5]). Surface-impact variables are of course particularly sensitive to the evolving surface observing network, such as changes in station density over land or the transition from ship to buoy observations over ocean, which feed the assimilation system. These factors underscore the need for sensitivity tests and cross-dataset comparisons when interpreting ERA5-based trends, especially outside the satellite era and in regions with limited observational coverage. However, at the present time ERA5 represents the best-suited product to investigate long-term variability and to perform attribution studies [2]. Of course, the temporal stability of reanalysis products such as ERA5 is not guaranteed, as discontinuities related to evolving assimilation systems and observational inputs can affect long-term consistency, which need further assessment for

future releases."

Yes, we used the latest release of ERA5 and we corrected the reference in the bibliography which was referring to the older version of the dataset.

**0.9**

For reproducibility, please also state how the teleconnection indices NAO etc. are computed or where they are retrieved from. Also for reproducibility and avoiding the need to look it up in (the various) methodological paper, please state how many analogues contribute in each case study to the factual and counterfactual composites, both in terms of time steps and distinct analogue cases. Also state if the geographical area in which analogues are taken from is moving in space or geographically fixed. The paper must be self-contained in its essential parts. With the sake of reproducibility, we added the following sentence indicating the source of the teleconnection indices, at line 192: "The monthly indices are computed from the NOAA/ERSSTv5 data and retrieved from KNMI's climate explorer. In particular, the ENSO index is the 3.4 version as defined by Huang et al. [3], the AMO index is computed as described in Trenberth and Shea [6] and the NAO index is the rotated empirical orthogonal function of Z500.". To clarify that the geographical area in which analogues are taken is fixed, we added the following sentence at line 169: "The analogues are searched in the same fixed area where we look for the SLP minimum.". The analogues which contribute in each case study to the factual and counterfactual composites are thirty, as mentioned at line XX "For both factual and counterfactual periods, we select the thirty best analogues". As mentioned in the answer to the general comment 0.3 we added two tables with the explicit analogues time-steps.

**0.10**

Figure 1 differences in lower row: It is very difficult to discern shading vs. no-shading. It could be interpreted as the value "0.0". Either use a mask or a different color (white, grey) than on the color bar to discern non-significant areas. We modified the figure as suggested, adding a white mask for the non-significant areas.

**0.11**

Figure 4 caption: Make even clearer that the data are for all dates (months) in the respective periods and the dot is for the actual event date (here 18.2.2022). According changes for similar plots in discussion of other cases. **We modified the captions as suggested.**

**0.12**

Discussion of North American Winter storm: It would be easier to start with the discussion of Figure 6, just for avoiding keeping scrolling back and forth from text to appendix figures. Or relocate Figure A7 to the main. We prefer to keep this sequence since we need firstly to highlight the meteorological features of all events which are unique for each event and then to show common features for turbulence-related metrics.

**0.13**

lines 340-344: repeat that this statement refers only to the analogues for Storm Eunice and the picture looks different for the European summer storm, and the North American and Typhoon examples. We modified the text as suggested.

**References**

- [1] Bill Bell, Hans Hersbach, Adrian Simmons, Paul Berrisford, Per Dahlgren, András Horányi, Joaquín Muñoz-Sabater, Julien Nicolas, Raluca Radu, Dinard Schepers, Cornel Soci, Sebastien Villaume, Jean-Raymond Bidlot, Leo Haimberger, Jack Woollen, Carlo Buontempo, and Jean-Noël Thépaut. The era5 global reanalysis: Preliminary extension to 1950. Quarterly Journal of the Royal Meteorological Society, 147(741):4186–4227, 2021.
- [2] Davide Faranda, Gabriele Messori, Erika Coppola, Tommaso Alberti, Mathieu Vrac, Flavio Pons, Pascal Yiou, Marion Saint Lu, Andreia NS Hisi, Patrick Brockmann, et al. Climameter: contextualizing extreme weather in a changing climate. Weather and Climate Dynamics, 5(3):959–983, 2024.
- [3] Boyin Huang, Peter W. Thorne, Viva F. Banzon, Tim Boyer, Gennady Chepurin, Jay H. Lawrimore, Matthew J. Menne, Thomas M. Smith, Russell S. Vose, and Huai-Min Zhang. Extended reconstructed sea surface temperature, version 5 (ersstv5): Upgrades, validations, and intercomparisons. *Journal of Climate*, 30(20):8179 8205, 2017.
- [4] A. J. Simmons. Trends in the tropospheric general circulation from 1979 to 2022. Weather and Climate Dynamics, 3(3):777–809, 2022.
- [5] Cornel Soci, Hans Hersbach, Adrian Simmons, Paul Poli, Bill Bell, Paul Berrisford, András Horányi, Joaquín Muñoz-Sabater, Julien Nicolas, Raluca Radu, Dinand Schepers, Sebastien Villaume, Leopold Haimberger, Jack Woollen, Carlo Buontempo, and Jean-Noël Thépaut. The era5 global reanalysis from 1940 to 2022. Quarterly Journal of the Royal Meteorological Society, 150(764):4014-4048, 2024.
- [6] Kevin E. Trenberth and Dennis J. Shea. Atlantic hurricanes and natural variability in 2005. *Geophysical Research Letters*, 33(12), 2006.